# Zinc Protoporphyrin-Rich Pork Liver Homogenates as Coloring Ingredients in Nitrite-Free Liver Pâtés

**DOI:** 10.3390/foods13040533

**Published:** 2024-02-09

**Authors:** Mar Llauger, Luis Guerrero, Jacint Arnau, Afra Morera, Jun-ichi Wakamatsu, José M. Lorenzo, Ricard Bou

**Affiliations:** 1Food Safety and Functionality Program, Institute of Agrifood Research and Technology (IRTA), Finca Camps i Armet s.n., 17121 Monells, Spain; 2Food Technology and Product Quality Program, Institute of Agrifood Research and Technology (IRTA), Finca Camps i Armet s.n., 17121 Monells, Spain; lluis.guerrero@irta.cat (L.G.); jacint.arnau@irta.cat (J.A.); 3Research Faculty of Agriculture, Hokkaido University, Kita-9 Nishi-9, Sapporo 060-8589, Japan; jwaka@agr.hokudai.ac.jp; 4Centro Tecnológico de la Carne de Galicia, Rúa Galicia Nº 4, Parque Tecnológico de Galicia, San Cibrao das Viñas, 32900 Ourense, Spain; jmlorenzo@ceteca.net; 5Área de Tecnología de los Alimentos, Facultad de Ciencias de Ourense, Universidade de Vigo, 32004 Ourense, Spain

**Keywords:** liver co-products, valorization, color, ingredient, clean label, non-nitrified liver pâtés

## Abstract

This study aimed to investigate the coloring ingredient potential of liver homogenates that form Zn protoporphyrin (ZnPP), a natural red pigment, after anaerobic incubation. Liver homogenates were used to develop nitrite-free sterile pork liver pâtés. These homogenates were applied in the formulation of pâtés directly or after centrifugation to obtain a pellet that was highly concentrated in ZnPP. Both the whole homogenate and its insoluble fraction were adjusted to pH 7.5 before their use in the formulation of pâtés with and without antioxidant (0.5% ascorbate plus 0.1% tocopherol) addition. Pâtés formulated with the whole homogenate showed color and texture characteristics that were similar to those of the positive control with nitrite. However, high levels of the insoluble fraction also led to pâtés with improved color characteristics but with a two-fold softened texture. Therefore, the form and amount of ZnPP added played roles in the final appearance of the product. The ZnPP pigment was more stable than heme in the sterilization treatment, and antioxidant addition proved to be unnecessary. The ZnPP-rich ingredients allowed for the preparation of nitrite-free cooked liver pâtés with a stable red color and could thus be potentially applied in other uncured cooked meat products.

## 1. Introduction

The color of meat and meat products plays a crucial role in attracting the attention of consumers and influencing their purchasing decisions [1]. The meat industry commonly uses nitrates and nitrites to give cured meat a distinctive color. Nitrates are reduced to nitrite by nitrate reductase-containing microorganisms. Nitrite is then converted to nitric oxide that subsequently reacts with myoglobin in meat and forms the nitrosylmyoglobin pigment. This pigment confers a reddish color to cured meat. Upon cooking, nitrite is denatured and converted into the pink pigment called nitrosyl hemochrome. Additionally, nitrite contributes to flavor development, prevents the oxidation of lipids and proteins, and inhibits the growth of pathogenic bacteria such as *Clostridium botulinum* [2]. However, the use of nitrates and nitrites has raised concerns regarding the formation of carcinogenic N-nitrosamines [3].

Alternatives for the use of pure nitrates and nitrites have gained interest for addressing the issue of N-nitrosamine formation and meeting consumer preferences for natural and clean-label products [4]. One strategy that has been extensively studied is the use of nitrate sources, such as plants or vegetables [5,6]. When combined with nitrate-reducing bacteria, these natural sources have been shown to facilitate color development in meat products to an extent similar to the addition of pure chemical sources [7,8]. The bacterial production of nitric oxide has also received attention for the reddening of meat products [9]. These approaches represent clean-label strategies but may require additional processing steps. Additionally, these approaches do not reduce the carcinogenic risk. In contrast, a strategy that avoids nitric oxide sources and subsequent N-nitrosamine carcinogenic risk involves the formation of the red pigment named zinc protoporphyrin (ZnPP).

The natural formation of ZnPP was demonstrated during the processing of dry-cured hams where it conferred a characteristic reddish color in ripened hams [10,11,12]. Notably, the long elaboration period of nitrite-free dry-cured ham allowed for the progressive formation of ZnPP. However, in many cooked meat products, the elaboration process takes several hours. Therefore, an important limitation for a broad number of applications is the slow formation of ZnPP in these products. Horse meat and porcine liver have shown higher ZnPP-forming capacities than that of pork, thereby opening new possibilities for nitrite-free meat product development [13,14]. Consequently, an optimized process for obtaining a potential coloring ingredient with preformed ZnPP was developed to valorize porcine liver [15]. This process involves the anaerobic incubation of pork liver homogenates (20% liver dispersed in an ascorbic acid and acetic acid aqueous solution) at pH 4.8 and 45 °C for 24 h. The resulting ingredient can be used in the formulation of several cooked products, such as liver pâté [16]. The safety of some cooked products can be guaranteed by inactivating sporulating microorganisms through sterilization. Liver pâté is often sold as a sterilized product and is therefore a good candidate for the study of an ingredient’s thermal stability and final product characteristics.

However, several aspects must be considered to ascertain the usefulness of liver homogenates as coloring agents for nitrite-free meat product development. For instance, when forming ZnPP, we found that the liver homogenate color after incubation was brown but exhibited a red color at pH 7 and above. For this reason, the pH was adjusted to 7.5 using pyrophosphates, which are common additives in pâté formulations to enhance the protein technological properties. However, the coloring capacity of the ingredient at a normal meat product pH needs to be examined. Other important aspects to consider are the effects of antioxidant addition and sterilization treatment on the ZnPP content and color. Antioxidant addition prevents pigment degradation and the generation of off-flavors in nitrite-free final products. However, the resulting ingredient is a partial liver autolysate with antioxidant properties [17]. Thus, it may be possible to take advantage of the presence of antioxidants. Water is normally added to the formulation of various meat products, such as pâté. Thus, the addition of water in the formulation can be omitted by replacing it with the whole homogenate after incubation. Alternatively, the ZnPP pigment, which remains insoluble at the end of the optimization process, could be concentrated 5-fold using centrifugation, thereby allowing for increased inclusion levels in meat product formulations. In reformulation processes, the replacement of liver with this ZnPP-rich ingredient (as a whole homogenate or in the form of pellets) may also affect the texture characteristics of the final product because the protein techno-functional properties may differ from those of the native liver. Finally, the interactions between ZnPP and other matrix compounds should not be disregarded, as free ZnPP binds to hemoglobin during the elaboration of hams [10]. Therefore, different ingredient addition methods, the ZnPP content and its interactions with other food components for optimal color development, and the impact of reformulation processes on the texture and lipid oxidation of nitrite-free cooked meat products remain to be studied. Despite the above, it is reasonable to hypothesize that the use of ZnPP-rich ingredients from porcine liver in meat products may offer a safe, eco-friendly, and healthy choice of natural and clean-label cooked meat products and ultimately provide an alternative to nitrites.

Therefore, this study aimed to investigate the effects of reformulation processes on the color, texture, and lipid oxidation of nitrite-free sterilized pâtés formulated with and without the addition of antioxidants. The reformulation processes involved the replacement of liver with two different liver homogenates that are rich in ZnPP. Additionally, the solubilization of ZnPP and its binding to hemoglobin were studied in different ZnPP-rich ingredient fractions, and the proximate composition, pH, instrumental color and stability, porphyrin pigment content, lipid oxidation, volatile profile, and textural properties of the pâtés were characterized.

## 2. Materials and Methods

### 2.1. Reagents and Standards

ZnPP, protoporphyrin IX (PPIX), and equine myoglobin were purchased from Merck (Darmstadt, Germany). Porcine chlorohemin (heme) was purchased from PanReac AppliChem (Barcelona, Spain). Acetonitrile suitable for high-performance liquid chromatography (HPLC) and trifluoroacetic acid (TFA) were purchased from VWR Chemicals (Radnor, PA, USA). Distilled and ultrapure water was obtained using a Milli-Q system (Millipore, Burlington, MA, USA).

### 2.2. Preparation of ZnPP-Rich Ingredients

Twenty porcine livers were purchased from a local slaughterhouse (Càrniques Juià S.A., Girona, Spain) on four different days (5 livers/day). Each day, the liver veins and connective tissues were trimmed; the livers were then diced and ground together at ≤4 °C in a meat cutter bowl to obtain four batches of liver paste, immediately vacuum-packed in aluminum bags, and then stored at −20 °C until needed.

ZnPP-rich ingredients were obtained from porcine liver homogenates, as described by Llauger et al. [15]. Briefly, liver homogenates consisting of 20% (*w*/*w*) fine liver paste were dispersed in an aqueous solution containing ascorbic acid at a final concentration of 1000 mg/L and acetic acid at 2500 mg/L and were adjusted to pH = 4.8 with 1 M NaOH. This mixture was incubated in a 5 L capacity reactor with a continuous flow of nitrogen at 45 °C for 24 h to obtain an autolyzed homogenate rich in ZnPP (ZnPP-H). Two different 5 L capacity productions of ZnPP-H were obtained from each batch and stored at −20 °C until use. Finally, to obtain a homogenous ingredient, one product from each batch was mixed to obtain two 20 L replicates.

Thus, ZnPP-H was used directly in the formulation of products as a potential coloring ingredient. However, the ZnPP pigment mainly remained in the insoluble fraction after centrifugation (5520× *g*, 20 min, at 4 °C). Hence, it was possible to obtain a pellet rich in ZnPP (ZnPP-P) that was also used in the formulation of products. The moisture content of ZnPP-P was 81%, which is similar to that of liver. The ZnPP-H ingredient was adjusted to pH 7.5 with tetrasodium pyrophosphate (TSPP) the day before pâté preparation to obtain a good initial color. In terms of the ZnPP-P ingredient, a saturated solution of TSPP was added at a level of 5% of the pellet weight, mixed, and adjusted to pH 7.5 before centrifugation under the same conditions to remove water and obtain the final ZnPP-P ingredient. After pH adjustment, the ingredients were kept refrigerated at 4 °C overnight.

### 2.3. Experimental Design and Pâté Preparation

The liver and fat used in pâté elaboration were bought at a local slaughterhouse (Càrniques Juià S.A., Girona, Spain), whereas the rest of the additives and spices were bought at Collelldevall S.L. (Banyoles, Spain). Eight experimental pâté formulations (2 kg each), including a positive control, negative control, and six formulations with ZnPP-rich ingredients, were prepared independently in duplicate in a pilot plant. The negative control formulation expressed per 100 g of prepared product was as follows: 31.2 g liver, 45 g pork back fat, 20 g water, 0.20 g black pepper, 1.6 g sodium chloride, and 2 g sodium caseinate. The same formulation was also used to elaborate the positive control with the addition of 0.015% sodium nitrite. Based on the negative control formulation, the other experimental pâté formulations were as follows: (1) 16% ZnPP-H, 16% replacement of the liver together with the complete replacement of water with ZnPP-H (i.e., 0% water + 26.2% liver + 25.0% ZnPPH of the total formulation); (2) 40% ZnPP-P, 40% replacement of the liver with ZnPP-P (i.e., 18.72% liver + 12.48% ZnPP-P of the formulation); and (3) 60% ZnPP-P, 60% replacement of the liver with ZnPP-P (i.e., 12.48% liver + 18.72% ZnPP-P of the formulation). The remaining formulations were prepared with the same ZnPP-rich ingredients, but the pâtés included 0.5% sodium ascorbate (E-301) and 0.1% tocopherol (E-309) as antioxidants (A), resulting in (4) 16% ZnPP-H/A, (5) 40% ZnPP-P/A, and (6) 60% ZnPP-P/A, respectively. Considering that the ZnPP-H was prepared from a 20% liver homogenate, the 16% ZnPP-H and 16% ZnPP-H/A formulations had the same theoretical protein content as that in the controls.

On the day of preparation, the liver and fat fractions were cut into cubes. The fat was scalded for 30 min in hot water (85 °C), and the ZnPP-rich ingredients were heated at 40 °C in a bath. Water, caseinate, and fat were then mixed in a cutter bowl. Subsequently, the liver was mixed with the rest of the ingredients according to the different formulations and emulsified while maintaining the temperature at ≥38 °C. Finally, the liver pâté mixture was mixed under a vacuum, manually distributed into 15 aluminum cans (7.3 Ø × 3.7 cm, ∼130 g pâté/can) until full, and then hermetically closed using a sealing machine (Talleres Ezquerra Seamers S.L., Navarra, Spain). The remaining liver pâté of each batch was immediately vacuum-packed in metalized bags (oxygen permeability of 1.5 mL/m^2^/24 h and low water vapor permeability of 1 g/m^2^/24 h) and stored at −80 °C until analysis. Finally, the pâté in the cans was sterilized in an autoclave at 112 °C for 50 min and cooled to 35 °C for 25 min. Liver pâtés were stored at room temperature, and three random cans of each formulation were selected for analysis 24 h after manufacture. One of the cans was used to measure the instrumental color, and the other two cans were used for texture analysis. The latter cans were placed in a storage cabinet at 25 °C overnight to ensure the samples had a homogeneous temperature. After these analyses, samples corresponding to the same formulation were mixed, and the pH was measured as described below. Then, samples were aliquoted, vacuum-packed in metalized bags, and stored at −20 °C until chemical analyses were carried out. Different formulations were replicated.

### 2.4. Physicochemical Analyses: Proximate Composition, pH, and Color

The moisture content was determined according to ISO 1442:1997 [18] until a constant weight was reached. The protein content was determined according to ISO 937:1978 [19] based on the Kjeldahl digestion method and multiplied by a factor of 6.25. The ash content was determined according to ISO 936:1996 [20]. Crude fat content was determined using the Soxhlet method (ISO 1443:1973 [21]). The chloride content was determined according to ISO 1841-2:1996 [22] using a potentiometric titrator (785 DMP Titrino, Metrohm AG, Herisau, Switzerland) and expressed as sodium chloride. The pH was determined using S40 SevenMulti (Mettler-Toledo SAE, Barcelona, Spain) and Inlab Solids Pro (Mettler-Toledo SAE) probes. All centesimal analyses were performed in duplicate, and six replicates were used to measure the pH value. Unless otherwise specified, the average of each formulation was considered a single measurement. Instrumental lightness (L*), redness (a*), and yellowness (b*) were measured according to the Commission International de l’Eclairage (CIE) CIELab space color on the inner surface of pâtés immediately after being cut crosswise (∼1.80 cm-thick slices), with three measurements per side using a Minolta Spectrophotometer CM-600d (Konica Minolta, Inc., Chiyoda, Tokyo, Japan). The illuminant was set to D65 with an observer angle of 10°. The color measurements were repeated on the same surface 15 min after being cut and exposed to atmospheric conditions under white fluorescent light (approximately 900 lx) to determine color stability.

### 2.5. Pâté Image Acquisition

High-quality images were acquired using a photographic system that included a calibrated Canon EOS 50D digital camera (Canon Inc., Tokyo, Japan) with a picture resolution of 15.1 megapixels and an objective Canon EF-S 18–200 mm f/3.5–5.6 IS. The camera was mounted in a black closet (1.06 × 106 × 2.50 m^3^) with halogen lights, Solux Q50MR16 CG/47/36°12 V/50 W/4700 K (Eiko Ltd., Shawnee, KS, USA). White balance was achieved using a white card (Lastolite Ltd., Leicestershire, UK). The camera was connected to a computer to store the images. Different pâté formulations were placed 50 cm below the camera on uniformly black surfaces. The canned products were photographed immediately after opening.

### 2.6. Textural Properties: Puncture and Spreadability Test

The puncture test was performed using the Texture Analyzer TA.HD Plus (Stable Micro Systems Ltd., Surrey, UK). Two cans of each formulation were tested with a cylindrical probe with a 5 mm diameter at a head speed of 1 mm·s^−1^. The force required to sink the probe to a depth of 20 mm was determined by using a load cell weighing 5 kg. Six measurements were performed for each can. The spreadability test was carried out using a Texture Analyzer TA.XT2 (Stable Micro Systems Ltd., Surrey, UK) with a spreader consisting of a conical tip submerging at a speed of 2 mm·s^−1^ with the shape of an inverted cone. Spreadability was expressed as the compression force required to deform the pâté by 75% using a load cell of 30 kg. Two measurements were performed for each can, and the average was considered a single measurement. All results were recorded using the Exponent software version 6.1.20.0 (Stable Micro Systems Ltd., Surrey, UK).

### 2.7. Determination of ZnPP, PPIX, and Heme Pigments

The ZnPP and PPIX content was determined similarly as described elsewhere [15]. Briefly, 2.5 g of pâté for the controls and 2.0 g for the remaining formulations were weighed in 50 mL centrifuge tubes and homogenized using an Ultra-Turrax T25 (IKA Werke GmbH & Co. KG, Staufen, Germany) for 30 s at 13,500 rpm in 10 mL of a cold solvent extraction mixture containing ethyl acetate/acetic acid/dimethyl sulfoxide (10:2:1, *v*/*v*/*v*) while the tube was immersed in ice. After a 20 min extraction at 4 °C to ease the fat removal, the extracts were centrifuged (2850× *g*, 20 min, 4 °C), the supernatant was filtered through filter paper (grade 1), and the filtrate was collected into amber glass tubes immersed on ice, and then finally transferred into 10 mL amber volumetric flasks. Subsequently, an aliquot (1 mL) was filtered through a 0.2 µm nylon syringe filter, and 200 µL was transferred to 96-microwell plates to read the ZnPP fluorescence with excitation set at 416 nm and emission at 588 nm, whereas PPIX fluorescence was read with an excitation set at 400 nm and emission set at 630 nm using a Varioskan Flash microplate reader (Thermo Fisher Scientific, Waltham, MA, USA).

The total heme pigment content was determined as described by Hornsey [23] with minor modifications. Briefly, 2.0 g of pâté was homogenized with 9 mL of 90% (*v*/*v*) aqueous acetone containing HCl (0.24 M) in triplicate. The samples were then gently magnetically stirred at 4 °C for 20 min. After centrifugation and filtration under the conditions mentioned above, an aliquot was filtered through a 0.2 µm nylon syringe filter and injected (30 µL) into an Agilent 1100 series HPLC system (Waldbronn, Germany) coupled to a UV-Vis detector set at 400 nm to quantify the total heme. The separation of porphyrins was performed on a Synergi column (150 × 4.6 mm, 4 µm, 80 Å) from Phenomenex (Torrance, CA, USA). The column elution was carried out at a flow rate of 1 mL/min at 35 °C using a gradient elution method with acetonitrile/water phases containing 0.05% TFA (20:80, respectively; initial eluting conditions), thereby increasing the proportion of acetonitrile to 100% within 10 min. After the run was completed, the column re-equilibration time was 5 min. All porphyrin results are expressed as micromoles per kilogram of liver on a dry weight basis.

### 2.8. Lipid Hydroperoxides

Lipid hydroperoxide content was determined by reduction with ferric thiocyanate, as described previously [24]. In duplicate, 3 g of pâté was weighed into a 50 mL Falcon tube, and then 5 mL of chloroform and 10 mL of methanol were added. The samples were then homogenized for 1 min at 9500 rpm while immersed in ice using an Ultra-Turrax T25 (IKA Werke GmbH & Co. KG, Staufen, Germany). Next, 5 mL of chloroform was added, followed by vortexing for 1 min, and 5 mL of water was added, followed by vortexing for 1 min. Then, the samples were centrifuged (3000× *g*, 30 min, at 4 °C), and after removing the upper phase (aqueous phase), the lipid extract (lower phase) was filtered through filter paper (grade 1) in a 10 mL volumetric flask. Finally, 200 µL of the lipid extract was mixed with 2.8 mL of methanol/butanol (2:1, *v*/*v*), incubated at room temperature for 20 min, and the absorbance was read at 500 nm. The results were expressed as µmols of cumene hydroperoxide equivalents per kg sample.

### 2.9. Hexanal Content and Volatile Profile

The volatile profiles of the pâtés were determined using headspace solid-phase microextraction (HS-SPME) with a Combi PAL injector autosampler (CTC Analytics, Zwingen, Switzerland). Before analysis, a 50/30 µm Divinylbenzene/Carboxen/Polydimethylsiloxane (DVB/CAR/PDMS) Stable Flex SPME fiber (Supelco, Bellefonte, PA, USA) was preconditioned at 260 °C for 30 min in the injector port of a gas chromatograph (GC) model 6850 coupled to a mass selective detector (MS) model, 5975C VL MSD (Agilent Technologies, Santa Clara, CA, USA). For HS-SPME extraction, 1 g of the pork liver pâté was weighed in a 10 mL amber vial, and 50 µL of 2-pentanone solution (Alfa Aesar, Ward Hill, MA, 99% purity) was added as an internal standard at a concentration of 60.49 µg/mL. The sample vials were incubated at 40 °C for 20 min, and the SPME fiber was inserted into the vial and exposed to the headspace for 40 min. The SPME fiber was desorbed and maintained in the injector port for 10 min at 280 °C in split mode (1:10). The compounds were separated in a DB-5MS capillary column (Agilent J&W Scientific; 30 m, 0.25 mm id, film thickness 1.00 µm). Helium was used as a carrier gas with a constant flow of 1 mL·min^−1^. The GC oven temperature program was started when the fiber was inserted into the GC injector port. The temperature was held at 40 °C for 10 min, ramped to 200 °C at 5 °C·min^−1^, and then to 280 °C at 80 °C·min^−1^, and held for 1 min, giving a total run time of 44 min. Mass spectral data were acquired in the range of 40–250 amu in scan acquisition mode, with the mas detector transfer line maintained at 280 °C and the mas source at 230 °C, while the mass quad was at 150 °C. Volatile compounds were tentatively identified by comparing their mass spectra with the database of the National Institute of Standards and Technology (NIST 2.0 version, Gaithersburg, MD, USA) and MassHunter software B.05.01. The results are expressed in area units per gram of sample.

### 2.10. Urea Polyacrylamide Gel Electrophoresis (PAGE) of Soluble Fractions

The interactions between ZnPP-H, ZnPP-P, and hemoglobin were examined by adding a hemolyzed red blood cell (RBC) fraction to different fractions of ZnPP-rich ingredients. The RBC fraction was obtained from commercial blood from a local slaughterhouse that contained tripolyphosphate solution (0.4%, *w*/*v*) as an anticoagulant. Blood was centrifuged at 2540× *g* for 15 min at 5–10 °C, and the cellular fraction was diluted 1:1 with MilliQ water. After 30 min under stirring, the RBC fraction was centrifuged at 20,900× *g* for 30 min at 15–20 °C to remove erythrocytic stroma.

The soluble fractions of ZnPP-H were obtained after adjusting the pH at 4.8, 6.7, and 7.5 with 1 M NaOH and centrifuging at 5520× *g* for 20 min at 4 °C. The resulting pellets at different pH levels were resuspended in the initial volume with distilled water and with a 5% RBC fraction aqueous solution. The ZnPP-P ingredient (pH 4.8) was resuspended at the initial volume with water, and the pH was adjusted to 4.8, 6.7, and 7.5. The same ingredients were resuspended in final 1% and 5% RBC aqueous solutions. When required, the pH of the resuspended samples was readjusted to the desired pH (4.8, 6.7, or 7.5), incubated at 4 °C for 1 h, and subsequently centrifuged under the same conditions. All supernatants containing the soluble fractions were filtered through filter paper, and the soluble ZnPP species at different pH values were separated using urea PAGE, as described previously [25]. Briefly, filtered supernatants were individually mixed in the following proportions: 29% sample, 50% sample buffer (50 mM Tris-HCl at pH 6.8 and 8 M urea), 20% glycerol, and 1% 2-mercaptoethanol. An aliquot of the RBC fraction was used as a marker and prepared as sample with 0.2% Coomassie Brilliant Blue G (CBB) prior to analysis. The hand-cast gel was prepared using a 4.5% stacking gel (4.5% acrylamide, 4 M urea, 125 mM Tris HCl, pH 6.8) and a 10% separating gel (10% acrylamide, 4 M urea, 375 mM Tris HCl, pH 8.8). The electrophoresis was run at 10 mA for 30 min followed by 20 mA for approximately 180 min at 4 °C. After electrophoresis, the gel was irradiated with a 420 nm purple light-emitting diode light source (OSSV5111A, OptoSupply Co. Ltd., Hong Kong, China). Fluorescent images were captured using a digital camera equipped with a 600 nm bandpass filter (BPB-60, Fujifilm Corp., Tokyo, Japan). To detect protein bands, the gel was stained with CBB solution (0.1% CBB, 40% methanol, 10% acetic acid) for 10 min and then destained with a solution containing 10% methanol and 7% acetic acid overnight.

### 2.11. Statistical Analysis

Statistical analyses were performed using XLSTAT statistical software (version Lumiver, 2023; New York, NY, USA). One-way analysis of variance (ANOVA) was used to examine whether there was a significant difference among the different pâté formulations in terms of proximate composition, chloride content, pH, instrumental color, porphyrin content, texture, and oxidation parameters. In terms of color, a series of one-way ANOVAs was performed for each exposure time (immediately after opening and after 15 min of exposure to air and light) to determine the existence of significant differences between the different formulations. Additionally, a series of one-way ANOVAs was performed for each formulation to determine the existence of significant differences between each exposure time. Likewise, a series of one-way ANOVAs was performed before and after the sterilization treatments to determine the existence of significant differences in porphyrin content between the different formulations. Additionally, a series of one-way ANOVAs was performed for each formulation to determine the existence of significant differences in porphyrin content as a result of the sterilization treatment (before and after). Significant differences among the different pâté formulations found using one-way ANOVA were evaluated using Tukey’s honest significant difference test. Statistical significance was set at *p* < 0.05.

## 3. Results

### 3.1. Proximate Composition, pH, and Color Characteristics

Table 1 shows the proximate content of the different pâté formulations. The moisture, fat, and NaCl contents were unaffected by the different formulations. However, a higher protein content was observed in the negative control and the 16% ZnPP-H and 16% ZnPP-H/A pâté formulations than in the 40% and 60% ZnPP-P/A pâtés, whereas the remaining formulations did not show differences among them. The ash content was the highest in the 16% ZnPP-H and 16% ZnPP-H/A pâté formulations and the lowest in the positive and negative controls, whereas the formulations with ZnPP-P with and without antioxidants had intermediate values. In general, high levels of ZnPP-rich ingredients resulted in pâtés with high pH levels. However, the pH levels of the pâtés with ZnPP-H were similar to those of the controls.

In terms of the pâté instrumental color measured immediately after the opening of the can, the lowest L* values were found in the 60% ZnPP-P pâté, whereas high values were recorded in the 16% ZnPP-H/A and positive control (Table 2). The highest a* value found in the positive control was attributed to the addition of nitrites and, consequently, to the formation of nitrosyl-heme. However, the lowest a* values found in the negative control were attributed to protein denaturation and the formation of hemichromes. Intermediate a* values were observed in the ZnPP-rich pâtés. However, the a* values in the 16% ZnPP-H and 16% ZnPP-H/A formulations were higher than those in the 60% ZnPP-P formulation, which, in turn, were higher than those in the remaining ZnPP-P formulations. In terms of b*, the lowest value was found in the positive control, followed by the negative control, whereas the highest values were found in the 16% ZnPP-H and 16% ZnPP-H/A formulations. Figure 1 demonstrates that the positive control and ZnPP-H pâtés were similar and redder than the other formulations, especially when compared with the negative control. In this latter formulation, the thermal treatment likely oxidized heme proteins.

The pâté color was measured after 15 min of exposure to air and light (Table 2). This exposure influenced all color parameters. In general, the samples darkened after air exposure, except for the negative control and the 40% ZnPP-P, 40% ZnPP-P/A, and 60% ZnPP-P formulations, which remained with similar values. In addition, except for the negative control, the pâtés’ redness decreased upon air exposure. Despite this decrease, the positive control and the 16% ZnPP-H and 16% ZnPP-H/A formulations remained redder than the other formulations after their exposure to air and light. Yellowness increased with air exposure in the control and the 40% and 60% ZnPP-P/A formulations, which may be indicative of oxidation processes [26].

### 3.2. Porphyrin Content before and after Sterilization Treatment

Table 3 shows the changes in pâté porphyrins (i.e., heme, ZnPP, and PPIX) after preparation and just before sterilization, and 24 h after thermal treatment. Before sterilization, the highest heme content was found in the positive control, followed by the 16% ZnPP-H and 16% ZnPP-H/A formulations and the negative control pâtés. The lowest heme content was observed in the 60% ZnPP-P/A formulation, which, in turn, was similar to that in the 60% ZnPP-P and 40% ZnPP-P/A formulations. In terms of ZnPP, the lowest content was observed in the positive and negative controls, whereas the highest content was observed in the 60% ZnPP-P/A formulation, followed by the 60% ZnPP-P, 40% ZnPP-P, and 40% ZnPP-P/A formulations. Thus, the addition levels and composition of ZnPP-rich ingredients determined the presence of ZnPP. Notably, ZnPP was more concentrated in ZnPP-P than in ZnPP-H. This is because ZnPP remains insoluble after 24 h of incubation at pH 4.8 and only approximately 1% of the ZnPP content in ZnPP-H is present in the soluble fraction after centrifugation at 5520× *g* for 20 min at 4 °C [15]. The precursor of ZnPP is PPIX [27]. Thus, the amount of PPIX was expected to depend on ZnPP. In agreement with this, the lowest PPIX content was found in the controls, and the highest was found in the 60% ZnPP-P and 60% ZnPP-P/A formulations. Additionally, a non-significant tendency towards a high ZnPP content and low heme content was observed in the pâté formulations with antioxidants when compared with that of their counterparts without antioxidants.

Examining the effect of sterilization revealed that thermal treatment mainly decreased the heme and ZnPP contents but did not affect the PPIX content (Table 3). However, the porphyrin content of the sterilized pâtés showed trends similar to those before the thermal treatment (Table 3). Thus, the heme content was high in the controls, and the inclusion of ZnPP-rich ingredients resulted in a decrease in the heme content. Simultaneously, the inclusion of ZnPP-rich ingredients increased the ZnPP content. Heme protein degradation by thermal treatment has been reported [28,29]. However, heme degradation was significant only in the presence of ZnPP-rich ingredients. In contrast, ZnPP degradation upon sterilization was only significant in the 40% and 60% ZnPP-P/A formulations.

### 3.3. Presence of Soluble ZnPP and Its Interaction with Hemoglobin

The presence of soluble ZnPP was examined after adjusting the pH of ZnPP-H to 4.8, 6.7, and 7.5. Figure 2A shows that only a small fluorescent band corresponding to ZnPP was present at the bottom of the gel when the pH was 7.5 (lane 4). The pellets obtained after the centrifugation of ZnPP-H at pH 4.8, 6.7, and 7.5 were resuspended to the initial volume with water or a 5% RBC fraction aqueous solution. Only the resuspension of the pellet with water at pH 7.5 resulted in the presence of soluble ZnPP at the bottom of the gel (Figure 2A, lane 7). However, when the pellets were resuspended with a 5% RBC fraction, it was possible to observe intense fluorescent bands at pH 6.7 and 7.5 (lanes 9 and 10) and very low intensity at pH 4.8 (lane 8), which are coincident with those of hemoglobin. A low-intensity band was also observed in the middle and bottom of the gel at pH 7.5 (lane 10). ZnPP-P was also resuspended in water and in 1% and 5% RBC fraction solutions (Figure 2B). Similar to ZnPP-H, when the pH was adjusted to 7.5, all ZnPP-P supernatants showed an intense fluorescence band at the bottom of the gel (lanes 4, 7, and 10). The same bands, but weak, also appeared at the bottom of the gel when resuspended with 1% and 5% RBC fractions at pH 6.7 (lanes 6 and 9). When ZnPP-P was added to a 5% RBC fraction solution and the mixture was adjusted to pH 4.8 and 6.7 (lanes 8 and 9), the resulting supernatants also showed a fluorescence band that corresponded to that of hemoglobin, as observed in Figure 2A. In addition, a low-intensity band was observed in the middle of the gel at pH 7.5 (lane 10).

### 3.4. Texture Properties: Puncture and Spreadability Tests

Two different tests were performed to characterize the pâté’s textural properties as it is a spreadable meat product. The highest hardness value in the puncture test was observed in the negative control, followed by the positive control (Table 1). The positive control exhibited similar hardness values to those of the 16% ZnPP-H/A formulation, which, in turn, was similar to that of the 16% ZnPP-H formulation. Moreover, the 40% and 60% ZnPP-P formulations had low hardness values.

The pâté spreadability showed trends similar to those of the puncture test (Table 1). The highest force was observed in the negative control group (*p* < 0.05). Low values were found for the positive control and the 16% ZnPP-H and 16% ZnPP-H/A formulations, which were similar between them. Significantly low values were found in the 40% and 60% ZnPP-P pâtés, which showed a tendency toward low values with increased liver replacement levels. In addition, the puncture and spreadability tests were unaffected by the addition of antioxidants (*p* > 0.05).

### 3.5. Lipid Oxidation and Volatile Profile

The effects of incorporating ZnPP-rich ingredients on the lipid hydroperoxide and hexanal contents are shown in Table 1. The lipid hydroperoxide content was similar in all formulations. The highest hexanal content was observed in the negative control, whereas the lowest content was observed in the 16% ZnPP-H and 16% ZnPP-H/A formulations. The positive control and the formulations containing ZnPP-P showed intermediate hexanal values. The addition of antioxidants did not affect the primary and secondary lipid oxidation products.

The volatile profiles are shown In Figure 3. Sixteen volatile compounds were tentatively identified in different pâtés. A similar volatile profile was found between the pâté formulations, except for the hexanal content, which was high in the negative control. In addition, other lipid-derived volatile compounds were identified in low abundance, such as 1-pentanol, 1-hexanol, and 1-octen 3-ol. The rest of the identified compounds were volatile terpenes, including α-thujene, α-pinene, sabinene, 2-β-pinene, β-myrcene, dl-limonene, β-thujene, and α-terpinolene, as monoterpenes, and linalool was identified as a terpenoid alcohol. Moreover, styrene, α-phellandrene, and α-humulene were identified.

## 4. Discussion

The decrease in the protein content with increased liver replacement percentages in the formulations could be due to the higher moisture and lower protein content of ZnPP-P (81.4% and 11.7%, respectively) compared with those of porcine livers (72–75% and 19–20%, respectively) [30,31]. The higher the concentration of ZnPP-rich ingredients, the higher the final pH. This finding can be attributed to the addition of TSPP to attain a pH of 7.5 in the ZnPP-rich ingredients. The higher ash contents of the 16% ZnPP-H and 16% ZnPP-H/A formulations compared with those of the ZnPP-P formulations could be due to the replacement of water with ZnPP-H containing dissolved salts. However, the pH range (from 6.63 to 6.87) was relatively narrow and similar to that reported in other studies [32,33]. Likewise, the proximate compositions of the different pâté formulations were within those described in the literature [32,33]. Thus, the pâté reformulation with ZnPP-rich ingredients did not substantially change the proximate compositions of the nitrite-free pâtés.

Conversely, there were important differences in the ZnPP and heme contents, which were mainly explained by the partial replacement of the liver with ZnPP-rich ingredients (Table 3). Nonetheless, in a previous study on liver homogenates, ascorbic acid addition caused a greater decrease in the heme content than the decrease without it, whereas the ZnPP content increased [15]. Hence, it is also possible that ZnPP formation may have occurred, to some extent, during pâté elaboration. This could explain not only the ZnPP content in the controls, but also the higher ZnPP content in the 60% ZnPP-P/A formulation compared with that in the 60% ZnPP-P formulation before sterilization. In general, the sterilization treatment caused a heme content decrease, which was significant in all formulations with ZnPP-rich ingredients, whereas the ZnPP decrease with the thermal treatment was only observed in the 40% ZnPP-P/A and 60% ZnPP-P/A formulations. This finding agrees with the reported increased stability of ZnPP extracts at thermal treatments up to 70 °C when compared with that of extracts containing nitrosyl-heme [34]. The contents of the different porphyrins explain the pâté appearances (Figure 1). The pinkish color of the positive control with nitrites was due to the presence of nitrosyl-heme, whereas the dull visual appearance of the negative control could be attributed to protein denaturation and hemichrome formation. Therefore, the red color of the pâtés containing ZnPP-rich ingredients can be attributed to the presence of ZnPP, which was relatively stable under the thermal treatment (Table 3). The stability of ZnPP during the sterilization treatments is of great interest for obtaining safe, red-colored, and nitrite-free meat products.

However, as shown in Figure 1 and Table 2 and Table 3, not only the ZnPP content (or level of addition) but also the form of the ZnPP-rich ingredients determined the red color of the product. Interestingly, ZnPP-H addition offered an intense red color (a* value) despite having a lower ZnPP concentration than that in ZnPP-P. The observed instrumental color (L*a*b*) values of the pâtés formulated with ZnPP-H were similar to those reported in other studies using different ingredients in the formulation of liver pâtés [26,33]. The effect of the ZnPP-H-soluble fraction on color was examined by replacing the water in the pâté formulation with the ZnPP-H supernatant obtained after centrifugation at pH 4.8 and subsequently adjusted to pH 7.5, causing little effect on the redness of the pâtés (Appendix A). It is unclear whether some soluble compounds in ZnPP-H interacted with other matrix compounds and participated in the appearance of the red color. However, in dry-cured hams, the binding of ZnPP to hemoglobin has recently been reported as the main coloring agent [10]. Therefore, we examined the presence of soluble ZnPP at different pH values when combined with or without hemoglobin. The results suggested that the pH and interaction of ZnPP with other compounds in the food matrix, such as hemoglobin, may have contributed to the increased content of soluble and bound ZnPP. The presence of soluble forms of ZnPP could explain the red appearance of the 16% ZnPP-H and 16% ZnPP-H/A formulations, which involved the mixing of insoluble ZnPP with a higher amount of hemoglobin from fresh liver than that in the other formulations. This fact is a consequence of the low percentage of liver replacement with ZnPP-rich ingredients in pâté formulations. The role of different ZnPP forms on color needs to be studied in more depth.

In addition to pigment thermal stability, the color stability under light and air exposure is another important feature to consider. In the negative control, air exposure had a very low blooming effect. In other formulations, porphyrin oxidation explained the observed color fading, which was measured as a loss of redness. In line with this finding, color fading has also been reported in nitrified meat products exposed to air and light, regardless of the addition of antioxidants such as ascorbate [35]. Under our conditions, ZnPP-H addition resulted in pâtés with redness and appearance similar to those of conventional nitrified pâtés, not only immediately after opening the cans but also after their exposure to light for 15 min. This indicates that the development of nitrite-free pâtés based on ZnPP is a good alternative to nitrite in terms of color and color stability.

In all formulations, the lipid hydroperoxide content was low compared to the values reported for pâtés and other cooked meat products [36,37,38]. Lipid hydroperoxides easily decompose into secondary lipid oxidation products at high temperatures, which explains their low content in pâtés after sterilization. Similar to our findings, the most abundant lipid oxidation breakdown compound reported in pâtés was hexanal [33,39]. The elevated hexanal content in the negative control can be partly explained by the relatively high content of heme iron, a well-known potent pro-oxidant [40]. Notably, the positive and negative controls contained similar amounts of heme before and after sterilization; however, heme was bound to nitric oxide in the positive control. Hence, the formation of nitrosyl-heme upon the addition of nitrifying agents reduces the catalytic effect of heme [41]. However, the heme content of the negative control was similar to that of the pâtés containing ZnPP-rich ingredients, whereas the hexanal content of the pâtés containing ZnPP-rich ingredients was lower than that of the negative control (Table 1 and Table 3). The decreased hexanal content in the pâtés with ZnPP-rich ingredients could partly be attributed to the tendency towards low heme amounts. Alternatively, the low hexanal values could be attributed to ascorbic acid addition and the formation of peptides with antioxidant properties during the ZnPP-rich ingredient preparation [17]. This latter explanation may be the main reason for the minimized lipid oxidation progression in the 16% ZnPP-H and 16% ZnPP-H/A formulations when compared with that of the positive control and some ZnPP-P pâté formulations. The antioxidant properties of ZnPP-H could also explain why further antioxidant addition to the ZnPP-H/A formulation seemed unnecessary to prevent oxidation. Likewise, the presence of antioxidant compounds in ZnPP-P should not be disregarded, given the lack of changes in ZnPP-P pâtés after antioxidant addition and the fact that the hexanal levels of these formulations were similar to those of the positive control with added nitrites.

Notably, the negative control showed higher hexanal values than those of the positive control, which could be explained by the well-known antioxidant properties of nitrite [41]. The progression of oxidation may have increased the hardness of the negative control by crosslinking the proteins with the oxidation products of lipids and/or proteins, resulting in hardened products during storage [42]. Compositional factors also greatly influenced the textural properties of the pâtés [43,44,45]. In this regard, the protein contents of the 16% ZnPP-H and 16% ZnPP-H/A formulations were similar to those of the positive and negative controls, thus explaining the observed similarities in the textural properties (Table 1). However, the protein contents in the 40% and 60% ZnPP-P formulations were also similar to those of the controls, and only the protein contents of the 40% and 60% ZnPP-P/A formulations were slightly lower than those of the negative control. Thus, the decrease in hardness values with the amount of ZnPP-rich ingredients can be attributed to partial protein denaturation caused by the decrease in pH during the elaboration of these ingredients and the reported proteolysis that occurred during their elaboration [17]. The degradation of native liver proteins into small proteins and peptides during the production of ZnPP-rich ingredients may explain the formation of a slightly intertwined gel network, resulting in weakened and spreadable gels. Moreover, proteolysis and a slightly lower protein content explained the more porous appearance in the formulations containing the ZnPP-P ingredient (Figure 1). Despite the observed changes in texture, they are not considered a limiting factor for the application of ZnPP-rich ingredients, because the texture can be tailored for the development of non-nitrified pâtés through formulation and structuring processes [46].

Finally, the volatile profiles agree with those observed in similar pâté formulations [33,39,47]. Volatile terpenes are relatively abundant and relevant because of their low threshold values that determine the aromatic characteristics of the final product. These compounds originate from spice and herb addition, which explains the absence of major differences between the different pâté formulations. Therefore, the addition of ZnPP-rich ingredients did not substantially change the pâté volatile profile when compared with that of the positive control with nitrite.

## 5. Conclusions

In summary, the type of ZnPP-rich ingredient influenced the color and texture characteristics of sterilized liver pâtés. The pâté formulations with ZnPP-H required low ZnPP-H addition levels, and they were more similar to the positive control containing nitrifying agents than to those formulated with ZnPP-P in terms of the redness and textural characteristics. However, increased ZnPP-P levels improved the redness and overall appearance. Hence, the ZnPP content, as well as the likely interaction between ZnPP and other matrix compounds (i.e., hemoglobin), contribute to improving the red color of the liver pâté. The ZnPP pigment was relatively stable against the thermal treatment, and the oxidation levels and volatile profiles were unaffected by the addition of ZnPP-rich ingredients. However, the omission of nitrifying agents without the incorporation of ZnPP-rich ingredients resulted in an undesirable color and high amounts of hexanal. In this regard, antioxidant addition was proven to be unnecessary to protect the liver pâté from lipid oxidation and pigment degradation during sterilization. Further research is needed to investigate the effects of ZnPP forms and their interactions on color, solubility, stability, taste, and lipid oxidation during pâté processing and storage to determine the shelf life of sterilized pâtés without nitrifying agents. Nonetheless, the incorporation of ZnPP-rich ingredients presents a promising strategy for manufacturing sterilized liver pâté with improved redness without the use of nitrifying agents. Finally, it is also necessary to study the safety and application of ZnPP-rich ingredients in other meat products to fully assess the potential of this strategy.

## Figures and Tables

**Figure 1 foods-13-00533-f001:**
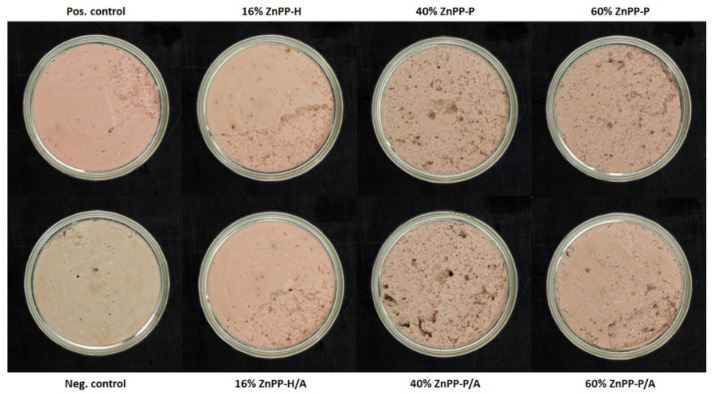
Photographs of canned pork liver pâtés after sterilization treatment and storage for 24 h at room temperature. The pâté formulations include the following: Pos. Control, positive control with nitrites; Neg. Control, negative control without nitrites; 16% ZnPP-H, complete replacement of water and 16% of the liver with whole autolyzed homogenate rich in Zn-protoporphyrin (ZnPP-H); 40% ZnPP-P and 60% ZnPP-P, replacement of 40% and 60% of the liver, respectively, with pellet rich in Zn protoporphyrin (ZnPP-P), which was obtained after centrifugation (5520× *g*, 20 min) of the whole autolyzed homogenate. All ZnPP-H and ZnPP-P pâté formulations in which the antioxidants sodium ascorbate and tocopherol were added (i.e., 16% ZnPP-H/A, 40% ZnPP-P/A, and 60% ZnPP-H/A) are also shown.

**Figure 2 foods-13-00533-f002:**
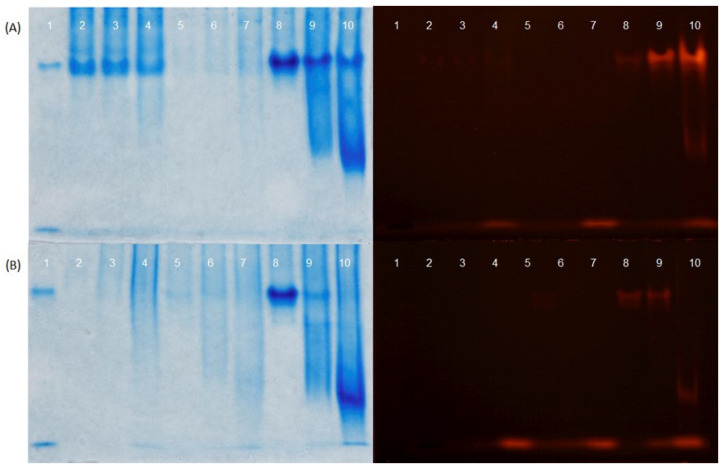
Coomassie Brilliant Blue-stained (left) and fluorescent (right) images of supernatants separated using urea polyacrylamide gel electrophoresis (PAGE). (**A**) Lane 1 corresponds to a hemolyzed red blood cell (RBC) fraction. Lanes 2, 3, and 4 correspond to soluble fractions of ZnPP-H obtained after centrifugation at 5520× *g* for 20 min at pH 4.8, 6.7, and 7.5, respectively. Lanes 5, 6, and 7 correspond to soluble fractions from the resulting ZnPP-H pellets obtained after centrifugation at pH 4.8, 6.7, and 7.5, respectively, which were resuspended at the initial volume with water and re-adjusted to the same pH. Lanes 8, 9, and 10 correspond to soluble fractions from the resulting ZnPP-H pellets obtained after centrifugation at pH 4.8, 6.7, and 7.5, respectively, which were resuspended at the initial volume with 5% RBC fraction solution and re-adjusted to the same pH. (**B**) Lane 1 corresponds to an RBC fraction. Lanes 2, 3, and 4 correspond to soluble fractions of ZnPP-P obtained after centrifugation at 5520× *g* for 20 min at pH 4.8, resuspended at the initial volume with water adjusted to pH 4.8, 6.7, and 7.5, respectively. Lanes 5, 6, and 7 correspond to soluble fractions of ZnPP-P resuspended at the initial volume with 1% RBC fraction aqueous solution adjusted to pH 4.8, 6.7, and 7.5, respectively. Lanes 8, 9, and 10 correspond to soluble fractions of ZnPP-P resuspended at the initial volume with 5% RBC fraction aqueous solution adjusted to pH 4.8, 6.7, and 7.5, respectively.

**Figure 3 foods-13-00533-f003:**
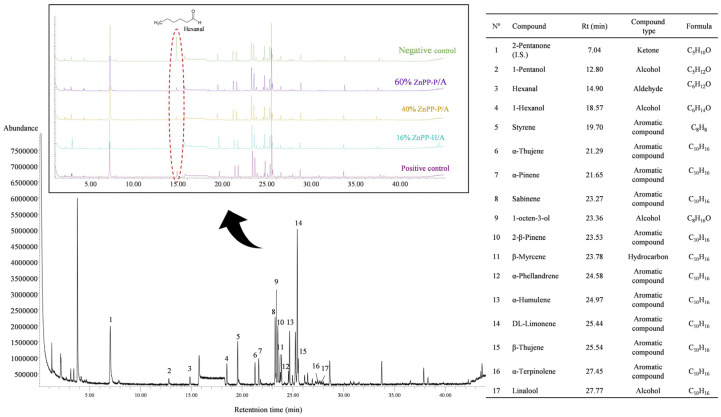
Total ion chromatogram of the volatile organic compounds found during the elaboration of canned pork liver pâtés containing ingredients rich in Zn-protoporphyrin. The compound number 1 (2-pentanone) was used as an internal standard.

**Table 1 foods-13-00533-t001:** Proximate composition, chloride content, pH, texture properties (hardness and spreadability), lipid hydroperoxides (LHP) content, and hexanal content of liver pâtés with added ZnPP-rich ingredients ^1^.

	Positive Control	Negative Control	16% ZnPP-H	40% ZnPP-P	60% ZnPP-P	16% ZnPP-H/A	40% ZnPP-P/A	60% ZnPP-P/A
Moisture (%)	53.7 ± 0.52	52.5 ± 2.50	53.1 ± 1.73	55.5 ± 1.10	55.9 ± 1.89	54.2 ± 0.42	54.1 ± 0.44	55.5 ± 1.35
Fat (%)	30.0 ± 1.39	30.1 ± 0.42	30.6 ± 1.98	28.3 ± 1.30	27.7 ± 1.47	29.2 ± 1.50	30.9 ± 2.61	29.3 ± 1.74
Protein (%)	10.0 ± 0.35 ^ab^	10.5 ± 0.24 ^a^	10.5 ± 0.31 ^a^	9.2 ± 0.15 ^ab^	9.3 ± 0.23 ^ab^	10.5 ± 0.24 ^a^	8.9 ± 0.26 ^b^	8.8 ± 0.56 ^b^
Ash (%)	2.03 ± 0.01 ^c^	2.03 ± 0.05 ^c^	2.86 ± 0.06 ^a^	2.29 ± 0.04 ^b^	2.43 ± 0.03 ^b^	2.85 ± 0.02 ^a^	2.32 ± 0.05 ^b^	2.43 ± 0.04 ^b^
NaCl (%)	1.65 ± 0.03	1.71 ± 0.05	1.70 ± 0.07	1.72 ± 0.13	1.67 ± 0.05	1.73 ± 0.05	1.74 ± 0.04	1.72 ± 0.10
pH	6.66 ± 0.01 ^c^	6.63 ± 0.03 ^c^	6.67 ± 0.02 ^c^	6.81 ± 0.01 ^ab^	6.87 ± 0.01 ^a^	6.68 ± 0.01 ^c^	6.77 ± 0.02 ^b^	6.87 ± 0.02 ^a^
Hardness (N)	2.0 ± 0.13 ^b^	2.3 ± 0.19 ^a^	1.8 ± 0.10 ^c^	1.2 ± 0.15 ^d^	0.8 ± 0.30 ^f^	1.9 ± 0.18 ^bc^	1.0 ± 0.18 ^e^	0.7 ± 0.06 ^f^
Spreadability (N)	9.8 ± 0.71 ^b^	11.5 ± 1.07 ^a^	8.8 ± 0.38 ^b^	6.5 ± 1.09 ^c^	4.5 ± 0.97 ^d^	9.4 ± 0.75 ^b^	5.4 ± 0.89 ^cd^	4.7 ± 0.33 ^d^
LHP (µmols CHP/kg)	16.3 ± 0.46	17.3 ± 2.91	9.2 ± 6.08	15.9 ± 1.39	17.4 ± 7.08	10.6 ± 6.73	20.4 ± 10.70	34.8 ± 25.89
Hexanal (AU·10^5^/g)	25.6 ± 1.94 ^b^	107.5 ± 13.12 ^a^	6.6 ± 1.92 ^c^	14.3 ± 6.38 ^bc^	10.8 ± 3.88 ^bc^	6.4 ± 2.62 ^c^	24.9 ± 8.49 ^b^	23.1 ± 7.57 ^b^

^1^ Results are shown as means ± standard deviation. Different letters in the same row (a–f) denote significant differences.

**Table 2 foods-13-00533-t002:** Instrumental color parameters (L*: lightness; a*: red/green; b*: yellow/blue) and color stability of liver pâtés with added ZnPP-rich ingredients ^1^.

Formulation	Initial Time (0 min)			After 15 min of Exposure to Light and Air		
	L*	a*	b*	L*	a*	b*
Positive Control	59.6 ± 0.79 ^a,x^	9.8 ± 0.27 ^a,x^	14.5 ± 0.24 ^e,y^	58.5 ± 0.82 ^a,y^	8.8 ± 0.24 ^a,y^	15.6 ± 0.29 ^d,x^
Negative Control	58.1 ± 1.05 ^ab^	1.4 ± 0.15 ^e,y^	15.5 ± 0.34 ^d,y^	58.1 ± 1.18 ^ab^	1.8 ± 0.19 ^e,x^	16.5 ± 0.37 ^c,x^
16% ZnPP-H	58.2 ± 0.76 ^ab,x^	7.5 ± 0.24 ^b,x^	18.9 ± 0.50 ^a^	56.9 ±1.13 ^bc,y^	6.6 ± 0.29 ^b,y^	19.1 ± 0.27 ^a^
40% ZnPP-P	57.1 ± 0.82 ^ab^	4.9 ± 0.23 ^d,x^	17.3 ± 0.48 ^c^	56.6 ± 0.92 ^c^	4.2 ± 0.35 ^d,y^	17.4 ± 0.51 ^b^
60% ZnPP-P	55.3 ± 1.60 ^c^	5.5 ± 0.34 ^c,x^	17.9 ± 0.34 ^b^	54.2 ± 1.75 ^d^	4.6 ± 0.33 ^c,y^	17.5 ± 0.75 ^b^
16% ZnPP-H/A	59.8 ± 1.10 ^a,x^	7.4 ± 0.29 ^b,x^	19.2 ± 0.38 ^a^	58.8 ± 0.79 ^a,y^	6.4 ± 0.38 ^b,y^	19.4 ± 0.52 ^a^
40% ZnPP-P/A	57.4 ± 1.24 ^ab^	4.6 ± 0.17 ^d,x^	16.9 ± 0.39 ^c,y^	57.6 ± 0.53 ^abc^	4.0 ± 0.21 ^d,y^	17.4 ± 0.34 ^b,x^
60% ZnPP-P/A	58.1 ± 0.92 ^ab,x^	4.7 ± 0.20 ^d,x^	17.1 ± 0.31 ^c,y^	57.0 ± 0.79 ^bc,y^	3.9 ± 0.12 ^d,y^	17.4 ± 0.19 ^b,x^

^1^ Results are means ± standard deviation. Different letters in the same column (a–e) denote significant differences between formulations, whereas different letters in the same row (x,y) denote significant differences when comparing the same color parameter (L*, a*, b*) before and after 15 min of exposure to atmospheric conditions and in the presence of light.

**Table 3 foods-13-00533-t003:** Heme, Zn-protoporphyrin (ZnPP), and protoporphyrin IX (PPIX) contents in liver pâté before (PB) and after the sterilization (SP) treatment ^1^.

Formulation	Heme (µmol/kg DM)		ZnPP (µmol/kg DM)		PPIX (µmol/kg DM)	
	PB	SP	PB	SP	PB	SP
Positive control	226 ± 20.3 ^a^	161 ± 30.6 ^a^	0.5 ± 0.11 ^c^	0.8 ± 0.11 ^c^	0.2 ± 0.03 ^d^	0.3 ± 0.05 ^d^
Negative control	193 ± 17.6 ^ab^	120 ± 28.9 ^ab^	0.4 ± 0.29 ^c^	1.9 ± 0.70 ^c^	0.2 ± 0.05 ^d^	0.4 ± 0.10 ^d^
16% ZnPP-H	198 ± 5.8 ^ab,x^	88 ± 8.4 ^b,y^	23.0 ± 4.64 ^c^	30.2 ± 2.35 ^c^	2.6 ± 0.59 ^d^	2.60 ± 1.17 ^cd^
40% ZnPP-P	181 ± 11.7 ^ab,x^	93 ± 3.3 ^ab,y^	122.7 ± 15.46 ^b^	76.6 ± 13.29 ^ab^	6.5 ± 0.83 ^c^	6.2 ± 2.00 ^bcd^
60% ZnPP-P	166 ± 10.0 ^b,x^	79 ± 5.8 ^b,y^	175.5 ± 18.69 ^b^	147.7 ± 7.88 ^a^	10.7 ± 0.95 ^ab^	10.1 ± 2.64 ^ab^
16% ZnPP-H/A	201 ± 14.5 ^ab,x^	78 ± 5.5 ^b,y^	29.2 ± 3.10 ^c^	28.1 ± 0.84 ^c^	3.0 ± 0.40 ^d^	2.8 ± 0.15 ^cd^
40% ZnPP-P/A	172 ±10.2 ^b,x^	90 ± 4.2 ^b,y^	166.6 ± 17.52 ^b,x^	85.1 ± 6.68 ^b,y^	8.5 ± 1.02 ^bc^	8.4 ± 0.24 ^abc^
60% ZnPP-P/A	156 ± 5.5 ^b,x^	75 ±7.3 ^b,y^	246.4 ±14.43 ^a,x^	132.2 ± 4.26 ^ab,y^	13.4 ± 0.89 ^a^	13.3 ±0.49 ^a^

^1^ Results are shown as means ± standard deviation. Different letters in the same column (a–d) denote significant differences between formulations, whereas different letters in the same row (x,y) denote significant differences when comparing the effect of the sterilization treatment.

## Data Availability

Data is contained within the article or Appendix A, further inquiries can be directed to the corresponding author.

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
