# Peer review of "Zinc Protoporphyrin-Rich Pork Liver Homogenates as Coloring Ingredients in Nitrite-Free Liver Pâtés"

_foods, 2024, doi:10.3390/foods13040533_

Round 1
Reviewer 1 Report
Comments and Suggestions for Authors
This manuscript explores the coloring potential of pork liver homogenates. This manuscript was well written. However, there were a few details that needed to be improved.
1. Fig.1: the canned pork liver pâtés adding ZnPP-H had many holes in it, what was the reason?
2. Table 1: the hardness of canned pork liver pâtés adding ZnPP-H was lower than the CK groups. Whether or not the product was sensory evaluated. How did it change in taste?
Author Response
Thank you for your comments.
Comment on Fig.1: the canned pork liver pâtés adding ZnPP-H had many holes in it, what was the reason?
In our opinion, this can be due to the slightly lowered protein content and the fact that the protein content of the ZnPP ingredient is more hydrolyzed. Following your comment, we decided to introduce a new sentence. See new version lines 617-618.
Comment on Table 1: the hardness of canned pork liver pâtés adding ZnPP-H was lower than the CK groups. Whether or not the product was sensory evaluated. How did it change in taste?
We measured instrumental hardness as described in the material and methods section. However, we also carried out a sensory analysis in this product but after 9 months storage. We would like to report these data in a publication dealing with pâté’s shelf life and the changes that occur during their storage at room temperature. In this publication, we will report that the sensory panel did not find differences in texture between the controls and pâtés with 16% level of addition. The panelists mainly found differences between samples in various color and texture descriptors whereas taste descriptors were almost unaffected.
Reviewer 2 Report
Comments and Suggestions for Authors
In this study, a new method was utilized to analyze the chemical composition of zinc protoporphyrin-rich pig liver homogenate and nitrite-free liver pate. Focusing on the impact of ZnPP on the quality and safety of liver pate, this study effectively combines detailed chemical analysis with practical significance in food science. It provides valuable insights into the moisture, protein, fat, and mineral content of liver pate. This methodology enhances our understanding of the nutritional value of this meat product, providing important insights for food safety and culinary applications. The manuscript is well-structured and thoroughly researched, emphasizing its importance in the fields of meat processing and food conservation. However, issues of organization, grammatical precision, and clarity of results and analysis need to be addressed.
Key areas of concern include:
Abstract: Ensure the abstract succinctly summarizes the key findings, including the properties of zinc-protoporphyrin (ZnPP) in meat sauce. It should also briefly mention the methodology for clarity.
Research Background: The introduction effectively illustrates the need for nitrite alternatives in meat products. However, extending previous research on ZnPP in meat products may strengthen this principle.
Design of experiment: Methods are detailed, but consideration should be given to specifying experimental variables and repeatability of results.
Methodology Details: Although the method is comprehensive, certain aspects such as details of control conditions, preparation steps, and analytical methods could be further detailed requiring clarity and reproducibility.
Results and Discussion: The results section is comprehensive. Be sure to directly relate these findings and hypotheses to existing literature, highlighting the novelty and significance of your findings.
Statistical analysis: Although statistical analysis is mentioned, a more detailed explanation of the choice of statistical tests and their relevance to the study would be beneficial.
Conclusion: The conclusion effectively summarizes the findings. Consider highlighting potential applications and future research directions to enhance its impact.
Practical Implications: Discuss the practical applications of your findings, particularly in the areas of food safety and consumer acceptance.
Grammar and language usage:
Abstract: Checking subject-verb agreement and tense agreement.
Introduction: Look at run-on sentences or overly complex sentence structures that may be simplified for clarity. Especially in the opening and the last few paragraphs of the introduction.
Materials and methods: Ensure that the use of passive voice is consistent and appropriate.
Discussion: Check To express complex scientific ideas clearly and avoid ambiguity. Position: Throughout the discussion, especially in the paragraphs explaining the data findings.
Conclusion: Ensure conciseness and clarity in summarizing the findings and avoid duplicating previous findings.
General: Troubleshoot common errors such as misplaced commas, inappropriate use of prepositions, and inconsistent punctuation. Location: Entire document.
Specific suggestions for improvements:
Assumptions and Objectives: Be Clear State the main assumptions and objectives at the beginning to guide the reader.
Literature review: Expand the scope of the literature review to include recent research and provide a more comprehensive background.
Methodology Details: Clarify any methodological ambiguities, especially in sample preparation and analytical procedures.
Data Interpretation: Strengthen interpretation of data by comparison with existing literature, and account for any deviations.
Practical Implications: Discuss the practical implications of your findings, particularly with regard to food safety and consumer acceptance.
Prospective Research: Suggest directions for future research based on your findings, especially in areas where your results were uncertain or unexpected.
Abstract (lines 14-27): Consider rephrasing for better flow and clarity. For example, "These liver homogenates can be used directly in the formulation of pate or as a centrifuged insoluble fraction highly concentrated in ZnPP" which may provide a clearer structure.
Introduction (lines 51-76): See for examples where sentence structure can be simplified for clarity. For example, “These methods represent clean label strategies but may require additional processing steps and do not reduce carcinogenicity risks” could be split into two sentences to improve readability.
Methodology section: Ensure consistency in the description of methods and procedures. Check for uniformity in tense and voice throughout this section.
Results Section: Pay attention to the consistency of tense when describing the findings. For example, "Pâtés formulated with the whole homogenate showed color and texture characteristics similar to those of the positive control with nitrite" could be checked for tense consistency in the context of the surrounding sentences.
Discussion and Conclusion Sections: In these sections, ensure that the conclusions drawn are clearly supported by the data presented. Revisit sentences that might seem overly complex or convoluted.
General Grammar and Punctuation: Throughout the document, look for minor grammatical errors like misplaced commas, incorrect preposition usage, and issues with subject-verb agreement.
In summary, the manuscript is well-structured and presents valuable research. Improvements in conciseness, minor grammatical adjustments, and a stronger emphasis on the implications and significance of the findings would enhance its quality. Additionally, incorporating visual elements like figures could make the paper more engaging for readers.
Comments on the Quality of English LanguageModerate editing of English language required
Author Response
Thank you for your comments.
Comments on key areas of concern:
Comment 1: Abstract: Ensure the abstract succinctly summarizes the key findings, including the properties of zinc-protoporphyrin (ZnPP) in meat sauce. It should also briefly mention the methodology for clarity.
We tried to follow your suggestion but, please, note there is a word limit of 200 words.
Comment 2: Research Background: The introduction effectively illustrates the need for nitrite alternatives in meat products. However, extending previous research on ZnPP in meat products may strengthen this principle.
We added a new citation dealing with nitrite free hams. It is worth mentioning that there are very few products with ZnPP other than hams. In fact, to the best of our knowledge, there are no other works in cooked meat products. These kinds of meat products require an almost instant formation of ZnPP. In paragraph 3, we explained part of our findings with the development of the ingredient with pre-formed ZnPP to be used as ingredient in the formulation of meat products. In paragraph 4, we explained the previous research and findings to strengthen the need of this study.
Comment 3: Design of experiment: Methods are detailed, but consideration should be given to specifying experimental variables and repeatability of results.
In our opinion, the material and methods section is sufficiently detailed. However, in case the reviewer considers that it is necessary to provide more information we would be happy to add and clarify any requested information.
Comment 4: Methodology Details: Although the method is comprehensive, certain aspects such as details of control conditions, preparation steps, and analytical methods could be further detailed requiring clarity and reproducibility.
We consider that the material and methods section is sufficiently detailed. However, we would be happy to clarify all the requested information that the reviewer considers necessary.
Comment 5: Results and Discussion: The results section is comprehensive. Be sure to directly relate these findings and hypotheses to existing literature, highlighting the novelty and significance of your findings.
This has been checked.
Comment 6: Statistical analysis: Although statistical analysis is mentioned, a more detailed explanation of the choice of statistical tests and their relevance to the study would be beneficial.
In our opinion, the statistical analysis section provides a good explanation of the purpose of each test.
Comment 6: Conclusion: The conclusion effectively summarizes the findings. Consider highlighting potential applications and future research directions to enhance its impact.
Following your recommendation, we added a new sentence at the end of the conclusions. Thank you.
Comment 6: Practical Implications: Discuss the practical applications of your findings, particularly in the areas of food safety and consumer acceptance.
We addressed that issue with the last sentence of the conclusions.
Comment 7: on grammar and language usage:
Abstract: Checking subject-verb agreement and tense agreement.
Introduction: Look at run-on sentences or overly complex sentence structures that may be simplified for clarity. Especially in the opening and the last few paragraphs of the introduction.
Materials and methods: Ensure that the use of passive voice is consistent and appropriate.
Discussion: Check To express complex scientific ideas clearly and avoid ambiguity. Position: Throughout the discussion, especially in the paragraphs explaining the data findings.
Ensure conciseness and clarity in summarizing the findings and avoid duplicating previous findings.
General: Troubleshoot common errors such as misplaced commas, inappropriate use of prepositions, and inconsistent punctuation. Location: Entire document.
We rewrote some sentences of the manuscript following your suggestions. We would like to inform that we can provide the invoice to prove that the English of the manuscript has been proof-read by professional services.
Comment 8: about specific suggestions for improvements:
Assumptions and Objectives: Be Clear State the main assumptions and objectives at the beginning to guide the reader.
Literature review: Expand the scope of the literature review to include recent research and provide a more comprehensive background.
Methodology Details: Clarify any methodological ambiguities, especially in sample preparation and analytical procedures.
Data Interpretation: Strengthen interpretation of data by comparison with existing literature, and account for any deviations.
Practical Implications: Discuss the practical implications of your findings, particularly with regard to food safety and consumer acceptance.
Prospective Research: Suggest directions for future research based on your findings, especially in areas where your results were uncertain or unexpected.
Thank you for your comments. The manuscript has been rewritten in some parts following your recommendations. We would like to emphasize that there are a limited number of studies with ZnPP and most of them deal with the elaboration of hams and to the best of our knowledge there are none in cooked meat products. For sure, it is the first publication in meat products with our developed ingredient.
Comment 9: Abstract (lines 14-27): Consider rephrasing for better flow and clarity. For example, "These liver homogenates can be used directly in the formulation of pate or as a centrifuged insoluble fraction highly concentrated in ZnPP" which may provide a clearer structure.
We followed your suggestion and modified this sentence.
Comment 10: Introduction (lines 51-76): See for examples where sentence structure can be simplified for clarity. For example, “These methods represent clean label strategies but may require additional processing steps and do not reduce carcinogenicity risks” could be split into two sentences to improve readability.
We followed your suggestion and modified this sentence.
Comment 11: Methodology section: Ensure consistency in the description of methods and procedures. Check for uniformity in tense and voice throughout this section.
This issue has been checked (new version lines 136-138). Thank you.
Comment 12: Results Section: Pay attention to the consistency of tense when describing the findings. For example, "Pâtés formulated with the whole homogenate showed color and texture characteristics similar to those of the positive control with nitrite" could be checked for tense consistency in the context of the surrounding sentences.
This sentence and others have been amended. We use past tenses to talk about our results whereas we prefer to use present tenses for well-known or proved facts. We would like to state that we paid for the services for proofreading the manuscript.
Comment 13: Discussion and Conclusion Sections: In these sections, ensure that the conclusions drawn are clearly supported by the data presented. Revisit sentences that might seem overly complex or convoluted.
In our opinion, the conclusions drawn are supported by our findings and we hypothesized that soluble forms of ZnPP may be behind the observed better red color of pâtes with 16% ZnPP-H.
Comment 14: General Grammar and Punctuation: Throughout the document, look for minor grammatical errors like misplaced commas, incorrect preposition usage, and issues with subject-verb agreement.
We revised the manuscript and corrected some errors. Thank you.
Comment 15: In summary, the manuscript is well-structured and presents valuable research. Improvements in conciseness, minor grammatical adjustments, and a stronger emphasis on the implications and significance of the findings would enhance its quality. Additionally, incorporating visual elements like figures could make the paper more engaging for readers.
Thank you for your comments. We tried to follow your recommendations but please note that we were requested to complete the review in a relatively short period of time to carefully address your general comments.

Reviewer 3 Report
Comments and Suggestions for Authors
The authors presented the results of development nitrite free liver pates by adding the liver homogenates. The idea of using liver homogenates as coloring agents and mechanism of forming ZnPP are properly described in the manuscript. The topic of this manuscript may be interesting not just for people from academia that deals with meat technology but also for professional from meat processing industry. Also, this paper may give good suggestion tot the researchers who will use the presented principle in some of the future research.
The layout of the paper is sufficiently good as well as the English language. The presented methodology is detailed and clear. My suggestion would be that formulation of negative control be presented in a table, as well as formulation of other samples, just for better visibility. In that way the readers could have even better insight in experimental set-up. The presented results back up the conclusions and they are given in a manner that can be easily understood.
Comments on the Quality of English Language
Minor editing of English language required.
Author Response
Thank you for your comments
With regards to reviewer's comment "My suggestion would be that formulation of negative control be presented in a table, as well as formulation of other samples, just for better visibility. In that way the readers could have even better insight in experimental set-up. The presented results back up the conclusions and they are given in a manner that can be easily understood"
We agree with the reviewer that this information could be easily understood in a table. However, we preferred to explain the formulations to avoid an excessive number of tables of figures. In our opinion, the description of the experimental design is clear and there is no room for misunderstandings. Moreover, we believe that the used abbreviations are self-explanatory and help to understand the experimental design.